REGISTERED REPORT PROTOCOL

# Investigating colonization patterns of the infant gut microbiome during the introduction of solid food and weaning from breastmilk: A cohort study protocol

Sara Dizzell[1], Jennifer C. Stearns[2,3], Jenifer Li[1,4], Niels van Best[5,6,7], Liene Bervoets[5], Monique Mommers[8], John Penders[5,7,9], Katherine M. Morrison[10,11], Eileen K. Hutton[1,4]*, on behalf of the GI-MDH Consortium Partners[¶]

1 Department of Obstetrics & Gynecology, McMaster University, Hamilton, Ontario, Canada, 2 Department of Medicine, McMaster University, Hamilton, Ontario, Canada, 3 Farncombe Family Digestive Health Research Institute, McMaster University, Hamilton, Ontario, Canada, 4 McMaster Midwifery Research Centre, McMaster University, Hamilton, Ontario, Canada, 5 School of Nutrition and Translational Research in Metabolism (NUTRIM), Department of Medical Microbiology, Maastricht University Medical Centre, Maastricht, The Netherlands, 6 Institute of Medical Microbiology, RWTH University Hospital Aachen, RWTH University, Aachen, Germany, 7 Vivo Planetary Health: An Affiliate of the World Universities Network (WUN), West New York, New Jersey, United States of America, 8 Department of Epidemiology, Care and Public Health Research Institute (CAPHRI), Maastricht University, Maastricht, The Netherlands, 9 School for Public Health and Primary Care (Caphri), Department of Medical Microbiology, Maastricht University Medical Centre, Maastricht, The Netherlands, 10 Department of Pediatrics, McMaster University, Hamilton, Ontario, Canada, 11 Centre for Metabolism, Obesity and Diabetes Research, McMaster University, Hamilton, Ontario, Canada

¶ Membership of the GI-MDH Consortium Partners is provided in the Acknowledgments. The lead author for this group is Dr. Eileen K Hutton.
* huttone@mcmaster.ca

This is a Registered Report and may have an associated publication; please check the article page on the journal site for any related articles.

## Abstract

The first exposures to microbes occur during infancy and it is suggested that this initial colonization influences the adult microbiota composition. Despite the important role that the gut microbiome may have in health outcomes later in life, the factors that influence its development during infancy and early childhood have not been characterized fully. Guidelines about the introduction of solid foods and cessation of breastfeeding, which is thought to have a significant role in the transition to a more adult-like microbiota, are not based on microbiome research. There is even less understanding of approaches used to transition to solid food in the preterm population. The purpose of this study is to identify the impact of early life dietary events on gut microbiome community structures and function among infants born at term and pre-term. We plan to prospectively monitor the gut microbiome of infants during two critical timepoints in microbial development: the introduction of solid foods and cessation from breastmilk. A total of 35 participants from three primary observational birth cohorts (two full-term cohorts and one pre-term cohort) will be enrolled in this sub-study. Participants will be asked to collect stool samples and fill out a study diary before, during and after the introduction of solids and again during weaning from breastmilk. We will use frequent fecal sampling analyzed using 16S rRNA gene profiling, metagenomics, metabolomics, and targeted bacterial culturing to identify and characterize the microbial communities, as well as provide

**Data Availability Statement:** All relevant data from this study will be made available upon study completion.

**Funding:** The Baby & Mi study was funded by grants from the Hamilton Academic Health Sciences Organization AFP Innovation Fund (received by KMM) and the Canadian Institutes of Health Research (received by EKH; reference #: MOP-136811). The Baby & Pre-Mi study and the Intensively Sampled sub-study were funded by a grant from the Canadian Institutes of Health Research (received by EKH; reference #: IMG-143923) through the European Union Joint Programming Initiative – A Healthy Diet for a Healthy Life. The LucKi Gut study was funded by a grant from The Netherlands Organisation for Health Research and Development (ZonMw) through the European Union Joint Programming Initiative – A Healthy Diet for a Healthy Life (Received by JP and MM; Project #: 529051010). The funders had and will not have a role in study design, data collection and analysis, decision to publish, or preparation of the manuscript. Links: https://www.hahso.ca/ https://cihr-irsc.gc.ca/e/193.html https://www.healthydietforhealthylife.eu/index.php/about https://www.zonmw.nl/en/.

**Competing interests:** The authors have declared that no competing interests exist.

insight into the phenotypic characteristics and functional capabilities of the microbes present during these transitional periods of infancy. This study will provide a comprehensive approach to detailing the effects of dietary transition from breastmilk to a more adult-like solid food diet on the microbiome and in doing so will contribute to evidence-based infant nutrition guidance.

## Introduction

The human gut contains the most diverse and dense microbiome in the body and is emerging as an important predictor of health and disease in humans [1, 2]. This complex ecosystem is the core of extensive research, and roles in metabolic function, digestion, physiology and immunological processes have been identified [3–5]. An altered or imbalanced microbiome has been associated with local gastrointestinal issues such as irritable bowel disease (IBD) and inflammatory bowel syndrome (IBS), and broader more systemic diseases such as obesity, type II diabetes, rheumatoid arthritis, and atopy [6–9]. The gut microbiome of healthy adults is considered to be relatively stable, however it may be transiently altered by perturbations such as antibiotic use, changes in diet, or infection [10–13]. In contrast, the infant gut microbiome is much less stable with fewer species and a higher proportion of *Bifidobacterium* [14, 15]. Despite the importance of a healthy gut microbiome for health, the factors that influence its development during infancy and early childhood have not been characterized fully.

Microbial changes that occur during infancy are influenced by numerous pre-natal, neonatal, and post-natal factors. These factors include, but are not limited to mode of delivery, gestational age, introduction of solid foods, breastmilk compared to formula feeding, weaning, geographical location, family member interactions and maternal diet [16, 17]. Exposure to the maternal vaginal and rectal microbiota during birth along with maternal skin and breastmilk allow for rapid colonization during the first hours and days of life [16, 18]. Because of both maturity of the infant system and vastly different neonatal experiences, these influences are likely to be different for infants born pre-term (prior to 37 weeks gestation). The pre-term infant is at increased risk for disruption in the normal development of the gut microbiome for a number of reasons including longer hospital stays, exposure to oral feeding at a more biologically immature state, higher proportion of formula feedings, and increased exposure to antibiotics. In combination with the delayed establishment of the gut microbiota in pre-term infants, it is suggested that the pre-term microbiota is less stable in comparison to full-term infants, which may contribute to the abundance of health issues observed in this population [19, 20]. Previous studies comparing the gut microbiota of pre-term and full-term infants have observed that there were fewer commensal anaerobes, such as *Bifidobacterium* and *Bacteroides*, and more opportunisitic pathogens such as Enterobacteriaceae and *Enterococcus* in pre-term infants [21–26].

Bacterial communities in the infant gut begin to resemble those of adults between 1 and 3 years of age [18, 27–31]. The ecological transition of the bacterial composition in the infant gut toward a more adult-like microbiota is thought to start around the time of introduction of solid foods, when a significant increase in microbial diversity and abundance is noted. The infant gut continues to progressively diversify until cessation from breastmilk when it becomes increasingly adult-like, as it is more stable and complex [27, 28, 30, 32–34]. Although food item selection during the introduction of solids may be an important contributing factor when considering microbial colonization in the infant gut, few studies have considered diet-related

colonization patterns [35]. Additionally, pre-term infants may be introduced to solid foods when their gut microbiota is at a more biologically immature state which would increase the risk of disrupting the normal microbiota development compared to full-term infants. Furthermore, cessation from breastmilk often occurs in conjunction with a more complex introduction of solid foods and formula or animal-based milk products. In comparison to early exclusive milk or formula feeding stages, the cessation of breastmilk is considerably less investigated [28].

There is little foundation from the field of microbiology for guidelines regarding introduction of solid foods and cessation of breastfeeding for term infants, and even less understanding of approaches used in the pre-term population. Insight is needed on the timing and importance of the potential deterministic factors that can serve as a target for manipulation. Therefore, this study aims to: identify the impact of early life dietary events on gut microbiome community structures and function among infants born at term and pre-term. At the time of introduction of solid food and cessation from breastmilk we will use frequent fecal sampling analyzed using 16S rRNA gene profiling, metagenomics, metabolomics, and targeted bacterial culturing to characterize the microbial communities in a subset of infants from three prospective birth cohorts. The use of culture-based methods in addition to molecular methods to profile microbial communities will allow us to isolate organisms and evaluate bacterial phenotypes *in vitro*, distinguish between live and dead cells, and enrich for less abundant bacteria using selective media [36]. The main objective of this study will be to describe the colonization patterns of the gut microbiome before, during, and after the introduction of solid food and before, during, and after weaning from breastmilk. We hypothesize that during the introduction to solid food and during the time around weaning from breastmilk we will see a shift in microbial composition to be more complex and less dominated by *Bifidobacterium*. We expect to see changes in the stool microbiome and metabolome within 14 days of each dietary change. However since little is known about how these dietary changes will impact the ecology of the gut microbiome, this work is largely hypothesis-generating rather than hypothesis-testing. This work will provide insight into timing and importance of the potential deterministic factors that can contribute to evidence-based infant nutrition guidance.

## Methods

### Study design

The Intensively Sampled sub-study is designed to prospectively monitor the gut microbiome of infants during two critical timepoints in microbial development: the introduction of solid foods and cessation from breastmilk. Participants from three primary observational birth cohorts, Baby & Mi, Baby & Pre-Mi (Hamilton, Ontario, Canada), and LucKi Gut (Maastricht, Netherlands) who consented to take part in the sub-study will be assessed for eligibility. The Baby & Mi and LucKi Gut cohorts are prospectively following full-term infants while the Baby & Pre-Mi cohort is following pre-term infants. All three of the primary cohorts are using similar questionnaires to collect information about the birth, infant diet, medications, and other exposures. In addition to participating in the primary study, eligible sub-study participants will be asked to provide data and collect stool samples over a 14 to 17-day intensive sampling period, which will occur before, during, and after the introduction of solid foods and again at the time of cessation from breastmilk. All three cohorts will participate in the introduction of solids phase. Full-term cohorts (Baby & Mi and LucKi Gut) will participate in the cessation from breastmilk phase of the study. The joint Hamilton Health Sciences–McMaster University Research Ethics Boards and Research Ethics Boards at all participating healthcare organizations approved the study in Canada. In the Netherlands, the study was approved by the

Medical Ethical Committee Maastricht University Medical Centre. Parental written informed consent was provided for all participants prior to enrollment in the study.

## Study population/participants/recruitment/sample size

A total of 35 participants will be enrolled for the Intensively Sampled sub-study. Twenty-five full-term infants will be recruited (15 from Baby & Mi and 10 from LucKi Gut) and ten pre-term infants will be recruited from the Baby & Pre-Mi cohort. This study focuses on within-subject differences in microbiota composition whereby each individual serves as its own control, an approach that increases power relative to studies comparing differences among individuals. With this study design, for an estimated expected mean of paired differences of 10 observed species and an expected standard deviation of the paired differences of 8, a sample size of 9 infants in each cohort is required to achieve a power of 80% and a level of significance of 5% (two sided) [37].

To be eligible for participation in the Intensively Sampled sub-study, participants must meet the criteria for the primary cohort (Table 1). Additionally, participants must meet the sub-study criteria (described below for each cohort). Any participants who consent and meet the eligibility criteria for the sub-study will be enrolled and given the study package containing the necessary materials, such as sample collection bags and the study dairy (for full study package materials list see S1 Appendix). For a descriptive summary of primary and Intensively Sampled sub-study details see Table 1.

Two sub-study cohorts of full-term babies will be formed: one in Canada (from the Baby & Mi study) and one in the Netherlands (from the LucKi Gut Study). The Baby & Mi study recruited 240 pregnant women from midwifery practices in Hamilton, Ontario and the surrounding area. Eligible participants included women who were planning to have a vaginal

**Table 1. Descriptive summary of the three cohorts that form the Intensively Sampled sub-study.**

|  | Baby & Mi Study | Baby & Pre-Mi Study | LucKi Gut Study |
|---|---|---|---|
| Study Location | Canada | Canada | The Netherlands |
| Sample Size | n = 15 | n = 10 | n = 10 |
| Source Population | Midwifery practices in Hamilton, Ontario and the surrounding areas | NICU at the McMaster Children's Hospital and NICU or Level 2 Nursery at St. Joseph's Healthcare in Hamilton, Ontario | Southern part of the province of Limburg, The Netherlands |
| Inclusion Criteria | Full-term (>37 weeks) | Pre-term (<37 weeks) | Full-term (>37 weeks) |
|  | Singleton | Twins or lower multitude birth | Singleton |
|  | Low risk (defined as being under the care of a midwife) |  | Low risk (defined as being followed though Baby Welfare Clinics) |
|  | Planning on having a vaginal birth |  |  |
| Exclusion Criteria | Caesarean section birth | Triplets or higher order multitude birth | Caesarean section birth intrapartum antibiotic administration |
|  | Admission to the neonatal or intensive care unit | Born with structural bowel abnormalities | Admission to the neonatal intensive care unit |
|  | Full weaning prior to the introduction of solid foods | Born with bowel disease that would require surgical intervention | Full weaning prior to introduction of solid food |
|  | Use of oral or IV antibiotics within 4 weeks of starting the study | Full weaning prior to the introduction of solid foods | Use of oral or IV antibiotics prior to introduction solid foods |
|  | The parent or guardian is unable to communicate in English | Use of oral or IV antibiotics within 4 weeks of starting the study | The parent or guardian is unable to communicate in Dutch |
| Follow-up time period | ~17-day sampling time period at the time of solid food introduction and again at the time of weaning from breastmilk | ~17-day sampling time period at the time of solid food introduction | ~14-day sampling time period at the time of solid food introduction and again at the time of weaning from breastmilk |

birth, considered to be low risk (defined as being under the care of a midwife), and were able to communicate in English in order to provide informed, signed consent. Women who had a pre-term birth (<37 weeks of completed gestation) and/or had a known multiple pregnancy were excluded. The LucKi Gut study is embedded within the Lucki Study and aims to recruit 300 pregnant women from the South Limburg area in the Netherlands via professionals involved in mother and child care and also through the internet (study website and Facebook). Women are eligible to participate in the LucKi Gut study if they give birth at >37 weeks of completed gestation. After giving birth, participants in the full term cohorts (LucKi Gut and Baby & Mi) who consented to the Intensively Sampled sub-study will be assessed for eligibility. Participants will be excluded if they gave birth by Caesarean section, if the infant is fully weaned prior to the introduction of solid foods, or the infant receives oral or intravenous (IV) antibiotics within 4 weeks of starting solid food consumption.

One sub-study cohort of pre-term infants will be formed from participants of the Baby & Pre-Mi study. Baby & Pre-Mi recruited 60 women within 72 hours of giving birth to a pre-term infant from the Neonatal Intensive Care Unit (NICU) at McMaster Children's Hospital or the Level 2 Nursery at St. Joseph's Healthcare in Hamilton, Ontario. Eligible participants included women who gave birth at <37 weeks gestational age and were able to communicate in English to provide informed, signed consent. Women were excluded if they had triplets or a higher order multiple birth, or if their baby was born with structural bowel abnormalities and/or were diagnosed with bowel disease that required surgical intervention. Participants in the primary Baby & Pre-Mi study who consented to the Intensively Sampled sub-study will be assessed for eligibility. Participants will be excluded if the infant is fully weaned prior to the introduction of solid foods and/or the infant receives oral or intravenous (IV) antibiotics within 4 weeks of starting solid foods.

## Sample/data collection

The Intensively Sampled sub-study consists of two main sampling frames; 1) the introduction of solid foods, and 2) weaning from breastmilk. Full-term infants will participate in both sampling frames. Because few pre-term infants experience exclusive breastmilk feedings, this cohort will not participate in the weaning from breastmilk sampling frame. Participants will be asked to collect stool samples and fill out the study diary before, during and after the introduction of solids and cessation from breastmilk. Participants will be given the study definition for introduction of solids as being "the time at which at least one feeding of breastmilk or formula is intentionally replaced with solid food". The study definition for cessation of breastmilk was "the last feeding of breastmilk".

Both Baby & Mi and Baby & Pre-Mi Intensively Sampled participants will collect samples over a 17-day period while the LucKi Intensively Sampled participants will collect samples over a 14-day period. For both parts of the study, participants will be instructed to collect samples on Days 1 to 3 as before introduction of solids/cessation from breastmilk, Day 4 as the day of introduction of solids/cessation from breastmilk and Days 5 to 14 or 17 as after introduction of solids/cessation from breastmilk. Additionally, the Baby & Mi and Baby & Pre-Mi Intensively Sampled participants will collect fresh stool samples Day 1 (for before) and Day 17 (for after). All other samples will be collected and stored frozen. A summary of the sample collection timepoints can be found in Fig 1.

For the collection of fresh stool samples, participants will be asked to call research staff immediately following the infant's bowel movement to arrange for sample pick-up and delivery. Additionally, parents will be instructed to add an anaerobic sachet to the sample collection bag along with the sample. The sample will then to be placed in a small cooler bag with an ice

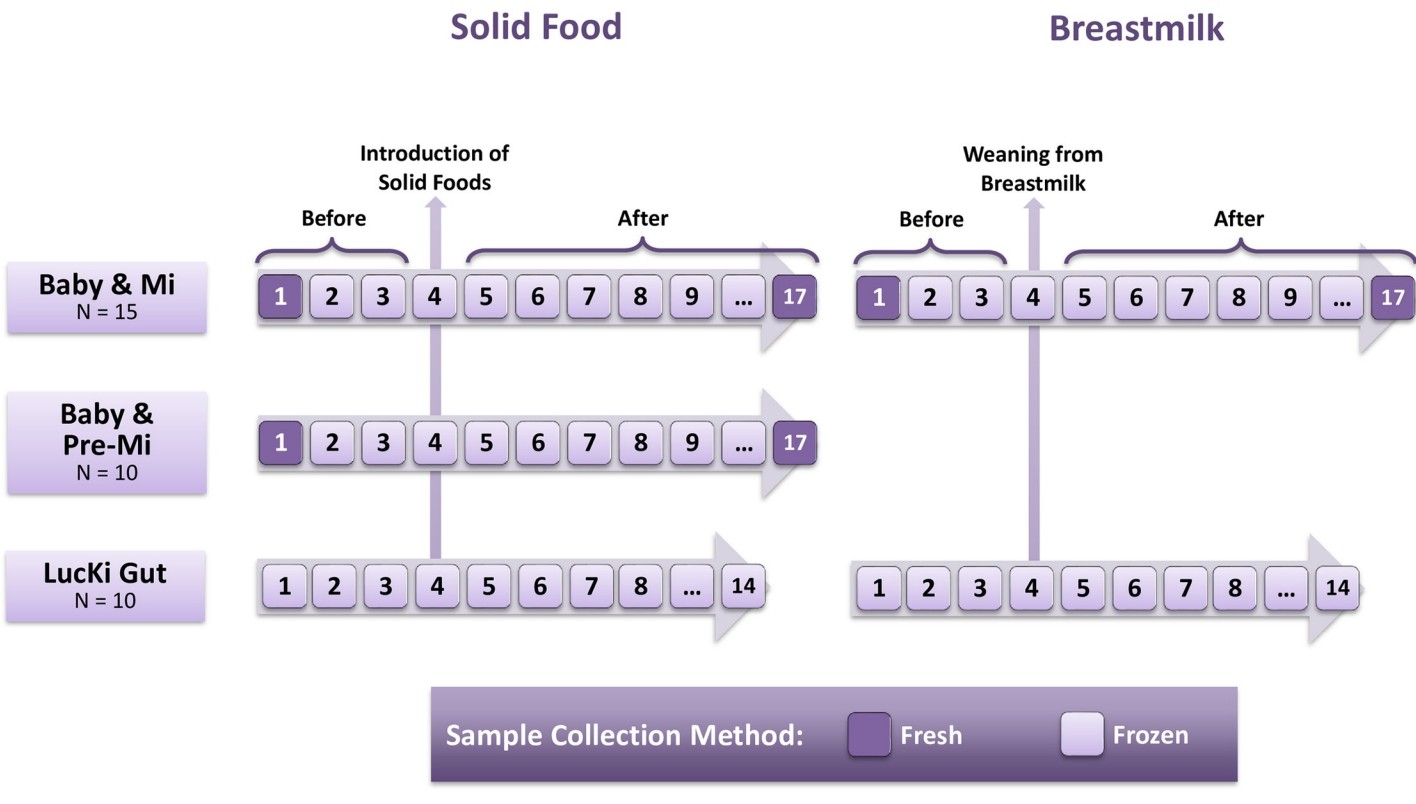

**Fig 1. Diagram of sample collection for the Intensively Sampled sub-study.**

pack to ensure that it remains anaerobic and cool during transport. The fresh samples will be processed in the laboratory within 4 hours of the bowel movement occurring. For the collection of frozen stool samples, participants will store the samples in their home freezers until all the samples are picked up or dropped off at the end of the sub-study period (Day 14 or 17).

In conjunction with collecting stool samples, participants will be asked to fill out a Study Diary every day during the intensively sampled period. On each day of study diary completion, parents will be asked to report if a stool sample was collected, the consistency of the stool sample based on the Bristol Stool Chart, if the sample came into contact with diaper cream, what type of diaper the sample was collected from, the number of bowel movements the infant had during the day, any medications the mother was on, how much time the infant spent sleeping, how many times the infant woke up during the night and how long the infant stayed awake if they woke up, and if the infant had any interaction with siblings, other children or animals. Additionally, all food consumption and medications will be recorded in an open text format. For full study diary details see S2 Appendix.

### Evaluation of the gut microbiome

To explore alterations in the infant gut during the introduction of solid foods and cessation from breastmilk, targeted bacterial culture, 16S rRNA gene sequencing, metagenomics, and metabolomics will be used.

**Outcome assessment.** Through frequent repeated sampling we aim to understand the daily dynamics in infant gut microbiota transition during two sentinel events which are

hypothesized to induce a critical disturbance to the infant gut microbiome: introduction of solid foods and weaning from breast milk. The primary outcome will be changes over time of the microbial communities as a whole following each of these events, in terms of bacterial membership, abundance and phylogeny, bacterial gene profiles and microbial metabolism, and strain phenotype changes of cultured bacterial strains over time. We expect that bacterial richness and diversity will increase over time and as the diet becomes more complex in both the full-term and the preterm infants. In the full-term cohort we expect there to be a loss of breastfeeding associated taxa such as some species of *Bifidobacterium*, that have been observed to decrease over the first year of life and to differ with breastfeeding status [38, 39]. We expect to describe significant shifts in the overall microbial community membership and substrate use over the study periods through prediction of bacterial genes and the abundance of metabolites in the stool that differ based on geography, birth mode and diet.

**16S rRNA gene sequencing.** The 16S rRNA gene is a highly conserved gene among bacterial species and other single celled organisms with specific variable regions. The assessment of genetic variation in these specific genomic regions, allows for identification, characterization and comparison of bacterial presence within a sample. To analyze genetic variation in specific genomic regions, highly targeted approaches such as amplicon sequencing were used. This method is well established, and provides extensive, in-depth information for identification, characterization and comparison of bacterial presence within a sample.

16S rRNA gene sequencing will be performed on fresh samples as well as a selection of frozen samples collected before, the day of, and after introduction of solid foods/cessation of breastmilk. 100 mg of stool will be used for DNA extraction from each sample. In cases where solid stool was not available, approximately 100 mg of stool will be collected from saturated diaper liner by sectioning the liner and subtracting the weight of the liner itself. DNA will be extracted from stool with established protocols as previously described [40]. Amplification of the bacterial 16S rRNA gene v3 region will be performed as previously described [41]. and Illumina libraries will be sequenced in the McMaster Genomics Facility with 250 bp sequencing in the forward and reverse directions on the Illumina MiSeq instrument. Raw sequencing data will be processed with standard software [42] and amplicon sequence variants (ASV) will be inferred using the DADA2 pipeline [43].

**Metagenomics.** Shotgun metagenomics will be performed on DNA extracted from fresh samples, described above, collected before and after the introduction of solid foods/weaning from breastmilk. Metagenomics sequences the entire genetic content of all organisms in a given sample and therefore not only gives information about the bacteria present, but also viruses, fungi, bacteriophages etc. Furthermore, metagenomics also provides information about the presence or absence of functional pathways within a sample which can give insight into phenotypic characteristics of organisms within the community. Metagenomics will be performed at the McMaster University Genome Centre (MUGC).

**Microbial culture.** Bacterial culturing was performed on fresh samples collected before and after the introduction of solid foods/weaning from breastmilk. Specific selective and differential media plates were chosen based on species of interest which were identified through a preliminary analysis of the Baby & Mi pilot cohort and were found to be persistent over the first year of life in the infant gut. Representative bacterial colonies will be picked from each media type and isolated. Additionally, plate pools of each media type will be collected. Detailed methods for bacterial culture are previously described by Lau et al., 2016 [44].

**Metabolomics.** The study of metabolomics gives insight into the functional capabilities of the gut microbes. It allows for the identification and quantification of molecular classes, compounds and metabolites produced by intestinal microbes. The gut microbiota is capable of modulating signalling pathways that regulate intestinal mucosa homeostasis through

metabolite production and fermentation. Three different metabolomic methods will be used on a subset of the samples collected during 14-day sampling period for both the introduction of solid foods and weaning from breastmilk. Both untargeted, direct infusion-mass spectrometry (DI-MS), and targeted, nuclear magnetic resonance spectroscopy (NMR) methods will be used to assess short chain fatty acids and other organic acids and alcohols. An additional targeted method, ultra performance liquid chromatography/multiple-reaction monitoring–mass spectrometry (UPLC/MRM-MS) will be used to assess bile acid profiles. Stool samples will be aliquoted and stored in a -80 freezer prior to being sent for metabolomic processing. All metabolomic methods will be performed at The Metabolomics Innovation Centre (TMIC) located at University of Alberta.

## Discussion

Many studies in the literature to date have taken a cross-sectional approach when assessing the microbiota of infants and thus have not informed the influence of exposures on intra-individual variation of the microbiome. In addition to the more recent longitudinal studies with wider intervals between sampling timepoints, our study will contribute to the literature by providing an understanding of the daily dynamics that occur in infant microbiota development around the time of dietary changes. In order to capture the potential change in microbiome as it occurs, in this study participants will collect daily stool samples (where possible) during the two sampling frames. A 2020 review of study design methods to evaluate the diet and microbiome relationship suggested that dense, longitudinal sampling with samples collected before, during and after an intervention of interest is ideal [45]. Our study will accomplish this by asking participants to collect daily fecal samples over a 14 to 17 day period that spans from before to after the introduction of solid foods and again at weaning. It has been previously reported that 7 to 9 sequential samples is ideal to account for intra-individual variation [45]. Consistent with this recommendation, we designed an approach that accounts for the possibility that our participants may not be able to achieve daily sampling for the entire sampling period.

Few studies have considered investigating the pre-term population specifically at the time of introduction of solid foods, which is an integral timepoint for colonization and development of the infant microbiota. Our study plans to carry out frequent sample collection, during windows of time that influence the development of the microbiome, such as the introduction of solid foods and cessation from breastmilk in order to provide more insight regarding microbial colonization and succession in full-term and pre-term infants.

Geographical location has been implicated as a relevant factor in infant gut microbiota colonization patterns. However, this is thought to be more related to dietary patterns and lifestyles in specific areas as regional diets and cultural practices of people living in similar ethno-geographic areas have been shown to have less inter-individual variation in their microbiomes [46]. Our international collaboration will allow us to look at differences and similarities between the gut microbiota of infants from Canada and The Netherlands and investigate the impact of cultural practices around food introduction and breastfeeding.

To identify the short-term changes to the microbial community function and host metabolism during weaning and introduction of solid food we will use molecular profiling, a targeted strategy for bacterial culturing, metagenomics and metabolomics. Previous studies have used metagenomic and metabolomic methods to assist in assessing functional diversity in the healthy adult microbiota [47–51]. The majority of metagenomic and metabolomic studies have focused on the adult microbiome, leaving the assessment of functional characteristics in the infant microbiome relatively unexplored. The use of metagenomic and metabolomic methods will give a better comprehension of the gut microbiome by not only giving information

regarding the composition and presence of microbes but also modulation and interaction between microbes and the host [52–54]. Despite the importance of functional processes in gut microbiota development, this area remains understudied as most literature to date uses molecular analyses based on the 16S rRNA gene sequencing which only examines the gut microbiota at taxonomic levels.

Until recently, it was thought that a large majority of bacteria within the gut were unculturable [55, 56]. The use of high throughput molecular based methods for evaluating bacterial communities within the gut is a preferential technique relied upon by majority of the scientific community. However, a recent study highlighted the importance of culture-enriched molecular profiling by showing that in comparison to culture-independent sequencing greater bacterial diversity was observed in the culture-enriched molecular profiling method [44]. Additionally, by selectively targeting bacterial families or species of interest, further studies can be performed to assess the functional roles of these important key players.

Given that the first exposures to microbes occur during infancy, it is suggested that this initial colonization is instrumental in influencing the adult microbiota composition, and subsequently may affect health outcomes later in life. In addition to microbiota colonization, establishment of constant dialogue between the immune system, metabolic pathways and the gut microbiome is developed. This provides further evidence, as noted in epidemiological studies, that host-microbe interactions instilled during infancy are major determinants of health or disease in adulthood [57–59]. Currently, the professional guidelines provided regarding the introduction of solid foods and cessation from breastmilk are varied among the governing bodies in different geographical locations which results in individual and cross-cultural variation. More importantly, these guidelines do not consider the effect of timing or type of solid food introduction on the infant gut microbiome. Our study will provide a comprehensive approach to detailing the effects of dietary transition from breast milk to a more adult-like solid food diet and in doing so will contribute to evidence-based infant nutrition guidance.

## Supporting information

**S1 Appendix. Study package materials list.**
(DOCX)

**S2 Appendix. Study diary.**
(PDF)

## Acknowledgments

GI-MDH Consortium Collaborators: Alison C. Holloway[1], Helen McDonald[2], Elyanne M. Ratcliffe[3,4], Jonathan D. Schertzer[4,5], Mike G. Surette[4,5], Lehana Thabane[6], Susanne Lau[7], Eckard Hamelmann[8]

[1] Department of Obstetrics and Gynecology, McMaster University, Hamilton, ON, Canada.

[2] McMaster Midwifery Research Centre, McMaster University, Hamilton, ON, Canada.

[3] Department of Pediatrics, McMaster University, ON, Canada.

[4] Farncombe Family Digestive Health Research Institute, McMaster University, Hamilton, Canada

[5] Department of Biochemistry & Biomedical Sciences, McMaster University, Hamilton, Canada

[6] Department of Clinical Epidemiology & Biostatistics, McMaster University, Hamilton, Canada.

[7] Department of Pediatric Pulmonology, Immunology and Intensive Care Medicine, Charité Universitätsmedizin Berlin, Germany.

[8] Children's Center Bethel, Protestant Hospital Bethel, University of Bielefeld, Germany.
The GI-MDH Consortium is led by Dr. Eileen Hutton (huttone@mcmaster.ca)

## Author Contributions

**Conceptualization:** Jennifer C. Stearns, Niels van Best, Liene Bervoets, Monique Mommers, John Penders, Katherine M. Morrison, Eileen K. Hutton.

**Funding acquisition:** Jennifer C. Stearns, Monique Mommers, John Penders, Katherine M. Morrison, Eileen K. Hutton.

**Methodology:** Sara Dizzell, Jennifer C. Stearns, Jenifer Li, Niels van Best, Liene Bervoets, Monique Mommers, John Penders, Katherine M. Morrison, Eileen K. Hutton.

**Writing – original draft:** Sara Dizzell.

**Writing – review & editing:** Jennifer C. Stearns, Jenifer Li, Katherine M. Morrison, Eileen K. Hutton.

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
