## [Decision Letter · Decision Letter 0]

8 Feb 2021

PONE-D-21-00254

Investigating colonization patterns of the infant gut microbiome during the introduction of solid food and weaning from breastmilk: a cohort study protocol

PLOS ONE

Dear Dr. Hutton,

Thank you for submitting your manuscript to PLOS ONE. After careful consideration, we feel that it has merit but does not fully meet PLOS ONE’s publication criteria as it currently stands. Therefore, we invite you to submit a revised version of the manuscript that addresses the points raised during the review process.

As pointed out by the reviewer, this protocol reports lacks a few important points:

- Is the sample size justified by power calculations from previous studies? There are enough studies that have followed longitudinally babies gut microbiome to perform such power calculations, or at least support what seems to be a relatively small number of participants.

- The manuscript would benefit from a "hypotheses and potential outcomes" section. It is difficult to understand why the study design, apart that it is (supposedly) different from what has been done before. 

- The authors state that daily sampling will be superior to detect fine changes. However, a case could be made that especially for babies/infants, significant stochasticity in microbiome dynamics is possible. This should at least be discussed.

We look forward to receiving your revised manuscript.

Kind regards,

Franck Carbonero, PhD

Academic Editor

PLOS ONE

Journal Requirements:

2. Please provide a sample size and power calculation in the Methods section, or discuss the reasons for not performing one before study initiation.

3. One of the noted authors is a group or consortium [GI-MDH Consortium Partners]. In addition to naming the author group, please list the individual authors and affiliations within this group in the acknowledgments section of your manuscript. Please also indicate clearly a lead author for this group along with a contact email address.

Additional Editor Comments:

As pointed out by the reviewer, this protocol reports lacks a few important points:

- Is the sample size justified by power calculations from previous studies? There are enough studies that have followed longitudinally babies gut microbiome to perform such power calculations, or at least support what seems to be a relatively small number of participants.

- The manuscript would benefit from a "hypotheses and potential outcomes" section. It is difficult to understand why the study design, apart that it is (supposedly) different from what has been done before.

- The authors state that daily sampling will be superior to detect fine changes. However, a case could be made that especially for babies/infants, significant stochasticity in microbiome dynamics is possible. This should at least be discussed.

Reviewers' comments:

Reviewer's Responses to Questions

**Comments to the Author**

1. Does the manuscript provide a valid rationale for the proposed study, with clearly identified and justified research questions?

Reviewer #1: Yes

2. Is the protocol technically sound and planned in a manner that will lead to a meaningful outcome and allow testing the stated hypotheses?

Reviewer #1: Partly

3. Is the methodology feasible and described in sufficient detail to allow the work to be replicable?

Reviewer #1: Yes

4. Have the authors described where all data underlying the findings will be made available when the study is complete?

Reviewer #1: No

5. Is the manuscript presented in an intelligible fashion and written in standard English?

Reviewer #1: Yes

6. Review Comments to the Author

You may also provide optional suggestions and comments to authors that they might find helpful in planning their study.

Reviewer #1: Overall:

No real hypothesis specified

Outcomes measures vague

Exact relationships between cessation of breastfeeding in full-term and pre-term infants and gut microbiome are not specified, therefore the study seems to be more of a simple observation than a hypothesis-testing scientific study.

Several grammatical errors and problems.

Lines 39-41 confusing, suggest re-writing

May need to hyphenate “full-term,” “adult-like,” and “evidence-based”

Line 52-need to indicate in the sentence that the study will detail effects of transition ON MICROBIOME

Line 66 need the word “and” after the word “species”

Line 79 need a comma before the word “and”

Line 81 need a comma after the word “infants”

Line 93, hyphenate “diet-related”

Consistently use “pre-term” or “preterm” or “pre term,” same goes for full term

Line 97-hyphenate “animal-based”

Line 100 hyphenate “culture-based”

Line 112, need the word “and” after “vitro”

Line 114, need comma after the word “during”

Line 124, remove the hyphen in “take part”

Sentence that spans line 140-142 needs to be edited for grammar, it does not make sense

Table 1 is confusing , should probably be edited down.

Table 1 row 2 should be the same in both the primary and IS cohorts columns

In table 1, there is a typo. Written is “Total Intensively Sample sub-study sample size…” it should say “Total Intensively Sampled…” this happens a few times

Multiple births at all are excluded from the full-term sub-sample cohorts, but it seems as though twins are allowed in the Baby&Pre-Mi cohort. Is this the case? If so, it shouldn’t be that way, and if not, the paragraph starting on line 167 needs to be edited to clarify.

Line 183 – why won’t pre-term infants be included in the weaning sampling frame? They should be

Line 188-192 -why are there different lengths of sample collection between baby+mi and baby+premi and lucki participants? Shouldn’t they be the same length?

Line 262- need to hyphenate “14-day” and also make clear that there are 14-day and 17-day sampling periods (or change protocol so that sampling periods are the same in all subsamples)

Outcome assessment – need to describe whether they believe outcomes will increase or decrease over time and based on what factors, and whether there will be differences between their subsamples and why.

7. PLOS authors have the option to publish the peer review history of their article (what does this mean?). If published, this will include your full peer review and any attached files.

Reviewer #1: No

---

## [Author Response · Author response to Decision Letter 0]

18 Feb 2021

Editor Comments:

As pointed out by the reviewer, this protocol reports lacks a few important points:

Is the sample size justified by power calculations from previous studies? There are enough studies that have followed longitudinally babies gut microbiome to perform such power calculations, or at least support what seems to be a relatively small number of participants. 

Thank you for pointing out this oversight. We have added the following statement to the methods section of our manuscript: 

“This study focuses on within-subject differences in microbiota composition whereby each individual serves as its own control, an approach that increases power relative to studies comparing differences among individuals. With this study design, for an estimated expected mean of paired differences of 10 observed species and an expected standard deviation of the paired differences of 8, a sample size of 9 infants in each cohort is required to achieve a power of 80% and a level of significance of 5% (two sided).” 

The manuscript would benefit from a "hypotheses and potential outcomes" section. It is difficult to understand why the study design, apart that it is (supposedly) different from what has been done before. 

We have added the following in our Introduction:

“We hypothesize that during the introduction to solid food and during the time around weaning from breastmilk we will see a shift in microbial composition to be more complex and less dominated by Bifidobacterium., We expect to see changes in the stool microbiome and metabolome within 14 days of each dietary change. Hhowever since little is known about how these dietary changes will impact the ecology of the gut microbiome, this work is largely hypothesis-generating rather than hypothesis-testing.”

We have also elaborated on our outcomes of interest in the Outcome Assessment section:

“ We expect that bacterial richness and diversity will increase over time and as the diet becomes more complex in both the full-term and the preterm infants. In the full-term cohort we expect there to be a loss of breastfeeding associated taxa such as some species of Bifidobacterium, that have been observed to decrease over the first year of life and to differ with breastfeeding status (Pannaraj et al., 2017 ; Henrich et, al., 2018 ). We expect to describe significant shifts in the overall microbial community membership and substrate use over the study periods through prediction of bacterial genes and the abundance of metabolites in the stool that differ based on geography, birth mode and diet.”

The authors state that daily sampling will be superior to detect fine changes. However, a case could be made that especially for babies/infants, significant stochasticity in microbiome dynamics is possible. This should at least be discussed. 

We recognize that that daily sampling may reveal stochasticity and that our methods may not necessarily be superior to wider sampling intervals. Instead, we think that both strategies are valuable and contribute different information to this field of research. We have modified what we had originally written in the Discussion to reflect this. In addition, we have provided a more in depth explanation for rationale for daily sampling in the Discussion and here:

In order to capture the potential change in microbiome as it occurs, we are asking participants to collect stool samples daily (where possible). A 2020 review of study design methods to evaluate the diet and microbiome relationship suggested that dense, longitudinal sampling with samples collected before, during and after an intervention of interest is ideal. We are attempting to accomplish both of these by asking participants to collect daily fecal samples over a 14 to 17 day period that spans from before to after the introduction of solid foods and again at weaning. We realize that it is quite possible that many of our participants will not be able to achieve daily sampling for the entire sampling period. The review states that up to 7 to 9 sequential samples is ideal in order to account for intra-individual variation and we feel that by striving to obtain samples over a 14 to 17 day period we will meet this target. 

Reviewers' comments:

1. Does the manuscript provide a valid rationale for the proposed study, with clearly identified and justified research questions?

Reviewer #1: Yes

2. Is the protocol technically sound and planned in a manner that will lead to a meaningful outcome and allow testing the stated hypotheses?

Reviewer #1: Partly

3. Is the methodology feasible and described in sufficient detail to allow the work to be replicable?

Reviewer #1: Yes

4. Have the authors described where all data underlying the findings will be made available when the study is complete?

Reviewer #1: No

5. Is the manuscript presented in an intelligible fashion and written in standard English?

Reviewer #1: Yes

6. Review Comments to the Author

You may also provide optional suggestions and comments to authors that they might find helpful in planning their study.

Reviewer #1: Overall:

No real hypothesis specified

Outcomes measures vague

Exact relationships between cessation of breastfeeding in full-term and pre-term infants and gut microbiome are not specified, therefore the study seems to be more of a simple observation than a hypothesis-testing scientific study.

Several grammatical errors and problems: 

Thank you for pointing out these details. We have made all of the suggested changes below.

Lines 39-41 confusing, suggest re-writing

May need to hyphenate “full-term,” “adult-like,” and “evidence-based”

Line 52-need to indicate in the sentence that the study will detail effects of transition ON MICROBIOME

Line 66 need the word “and” after the word “species”

Line 79 need a comma before the word “and” 

Line 81 need a comma after the word “infants”

Line 93, hyphenate “diet-related”

Consistently use “pre-term” or “preterm” or “pre term,” same goes for full term

Line 97-hyphenate “animal-based”

Line 100 hyphenate “culture-based”

Line 112, need the word “and” after “vitro” 

Line 114, need comma after the word “during” 

Line 124, remove the hyphen in “take part”

Sentence that spans line 140-142 needs to be edited for grammar, it does not make sense

Table 1 is confusing , should probably be edited down.

Table 1 row 2 should be the same in both the primary and IS cohorts columns

In table 1, there is a typo. Written is “Total Intensively Sample sub-study sample size…” it should say “Total Intensively Sampled…” this happens a few times

Line 262- need to hyphenate “14-day” and also make clear that there are 14-day and 17-day sampling periods (or change protocol so that sampling periods are the same in all subsamples)

Other issues:

Multiple births at all are excluded from the full-term sub-sample cohorts, but it seems as though twins are allowed in the Baby&Pre-Mi cohort. Is this the case? If so, it shouldn’t be that way, and if not, the paragraph starting on line 167 needs to be edited to clarify. 

Multiple gestations were excluded from the full term cohorts. The preterm cohort did include multiple gestations (twins). These were the eligibility criteria of the source cohorts (LucKi, Baby & Mi and Baby & Pre-Mi) which were chosen for clinical reasons relevant to the primary objectives of the source cohorts. We could have excluded multiple gestations from this intensively sampled subset of preterm infants but chose not to in order to maximize the number of participants that would be eligible for this sub-study. We recognize that the inclusion of twin participants reduces their statistical independence however, the collection of samples before, during and after each dietary change allows each infant to serve as their own baseline comparison. This follows the methodology of previous high impact studies of preterm infants which included multiple gestations (Xiong et al, 2017 and Raveh-Sadka et al, 2016).

Xiong W, Brown CT, Morowitz MJ, Banfield JF, Hettich RL. Genome-resolved metaproteomic characterization of preterm infant gut microbiota development reveals species-specific metabolic shifts and variabilities during early life. Microbiome. 2017 Dec;5(1):1-3.

Raveh-Sadka T, Firek B, Sharon I, Baker R, Brown CT, Thomas BC, Morowitz MJ, Banfield JF. Evidence for persistent and shared bacterial strains against a background of largely unique gut colonization in hospitalized premature infants. The ISME journal. 2016 Dec;10(12):2817-30.

Line 183 – why won’t pre-term infants be included in the weaning sampling frame? They should be. 

Preterm infants will not be included in the weaning sampling time frame because breastfeeding rates in the preterm population are lower than among full terms. By sampling from only those that breastfed we would be creating a biased cohort and our findings would not be generalizable to the general preterm population. We have clarified this in the manuscript.

Line 188-192 -why are there different lengths of sample collection between baby+mi and baby+premi and lucki participants? Shouldn’t they be the same length?

This study is a collaboration between three existing cohorts. As a result of varying resources and situations, the duration of sample collection between cohorts differs. We do not feel that this will impact our ability to compare results between cohorts.

Outcome assessment – need to describe whether they believe outcomes will increase or decrease over time and based on what factors, and whether there will be differences between their subsamples and why.

See reply above to the request for a “Hypothesis and Outcome” section. 

7. PLOS authors have the option to publish the peer review history of their article (what does this mean?). If published, this will include your full peer review and any attached files.

Do you want your identity to be public for this peer review? For information about this choice, including consent withdrawal, please see our Privacy Policy.

Reviewer #1: No

---

## [Editor Report · Decision Letter 1]

9 Mar 2021

Investigating colonization patterns of the infant gut microbiome during the introduction of solid food and weaning from breastmilk: a cohort study protocol

PONE-D-21-00254R1

Dear Dr. Hutton,

We’re pleased to inform you that your manuscript has been judged scientifically suitable for publication and will be formally accepted for publication once it meets all outstanding technical requirements.

Kind regards,

Franck Carbonero, PhD

Academic Editor

PLOS ONE
---

## [Editor Report · Acceptance letter]

26 Mar 2021

PONE-D-21-00254R1 

Investigating colonization patterns of the infant gut microbiome during the introduction of solid food and weaning from breastmilk: a cohort study protocol 

Dear Dr. Hutton:

I'm pleased to inform you that your manuscript has been deemed suitable for publication in PLOS ONE. Congratulations! Your manuscript is now with our production department. 

Kind regards, 

on behalf of

Dr. Franck Carbonero 

Academic Editor

PLOS ONE